# COVALENCE STUDY: Immunogenicity and Reactogenicity of a COVID-19 mRNA Vaccine in an Open-Label Cohort of Long-Survivor Patients with Metastatic Lung Cancer

**DOI:** 10.3390/vaccines13030273

**Published:** 2025-03-05

**Authors:** Emanuele Vita, Federico Monaca, Luca Mastrantoni, Geny Piro, Giacomo Moretti, Ileana Sparagna, Alessio Stefani, Antonio Vitale, Giovanni Trovato, Mariantonietta Di Salvatore, Maurizio Sanguinetti, Andrea Urbani, Luca Richeldi, Carmine Carbone, Emilio Bria, Giampaolo Tortora

**Affiliations:** 1UOSD Oncologia Toraco-Polmonare, Comprehensive Cancer Center, Fondazione Policlinico Universitario A. Gemelli IRCCS, Università Cattolica del Sacro Cuore, 00168 Rome, Italy; mariantonietta.disalvatore@policlinicogemelli.it (M.D.S.); emilio.bria@unicatt.it (E.B.); 2UOC Oncologia Medica, Comprehensive Cancer Center, Fondazione Policlinico Universitario A. Gemelli IRCCS, Università Cattolica del Sacro Cuore, 00168 Rome, Italy; federico.monaca@guest.policlinicogemelli.it (F.M.); geny.piro@policlinicogemelli.it (G.P.); ileana.sparagna@guest.policlinicogemelli.it (I.S.); alessio.stefani@unicatt.it (A.S.); antonio.vitale@guest.policlinicogemelli.it (A.V.); giovanni.trovato@guest.policlinicogemelli.it (G.T.); carmine.carbone@policlinicogemelli.it (C.C.); giampaolo.tortora@policlinicogemelli.it (G.T.); 3UOC Chimica, Biochimica e Biologia Molecolare Clinica, Fondazione Policlinico Universitario A. Gemelli IRCCS, Università Cattolica del Sacro Cuore, 00168 Rome, Italy; giacomo.moretti@policlinicogemelli.it (G.M.); andrea.urbani@policlinicogemelli.it (A.U.); 4UOC Microbiologia, Fondazione Policlinico Universitario A. Gemelli IRCCS, Università Cattolica del Sacro Cuore, 00168 Rome, Italy; maurizio.sanguinetti@policlinicogemelli.it; 5UOC Pneumologia, Fondazione Policlinico Universitario A. Gemelli IRCCS, Università Cattolica del Sacro Cuore, 00168 Rome, Italy; luca.richeldi@policlinicogemelli.it

**Keywords:** COVID-19 vaccine, lung cancer, immunotherapy

## Abstract

**Background:** As COVID-19 has become an epidemic, we conducted an open-label study aimed to identify immunogenicity and reactogenicity of boosters of the BNT162b2 vaccine in a real-world cohort of long-survivor metastatic lung cancer patients (LS-mLC pts). **Methods and Analysis:** According to the timing of the booster dose (BD) and SARS-CoV-2 infection (Cov-I) during anticancer treatment (ACT), between October 2021 and February 2022, we prospectively enrolled 166 cancer patients into five parallel cohorts. The primary endpoints were seroprevalence of IgG Anti-spike-RBD (anti-S IgG) at two pre-defined timepoints (T1: +30–90 days after BD; T2: +6 months +/− 4 weeks after BD). As an exploratory endpoint, we compared the median pre-vaccination value of four cytokines (IL-6, IL-2R, TNF-α, IL-10) with post-BD values in immunotherapy-treated pts (IO-pts). **Results:** The anti-S IgG seropositivity rate was 100% at T1 and 98.8% at T2. After 6 months, hybrid immunisation was associated with a higher median anti-S IgG titre compared to vaccine-alone-induced seroconversion (*p* < 0.0001). In uninfected pts, the median anti-S IgG titre was significantly lower in IO-pts compared to non-IO-pts (*p* = 0.02); no difference was found when comparing myelosuppressive or not ACT. Among the 68 IO-pts, 5 pts (7.3%) showed a significant increase (≥1.5 fold) of at least two cytokines in post-BD samples, without reporting ir-AEs. **Conclusions:** Boosters of the COVID-19 mRNA vaccine were effective and safe. In IO-pts without recent Cov-I, additional BDs should be considered to prolong serological immunity.

## 1. Introduction

During the COVID-19 pandemic, vaccination has provided an unprecedented opportunity to decrease morbidity and mortality in cancer patients who have increased fatality risk after SARS-CoV-2 infection. Particularly, lung cancer patients (LC pts) reported a higher prevalence of severe disease and poorer outcomes because of their underlying malignancy, smoking habits, oncological therapy and pre-existing cardiopulmonary comorbidities [1,2,3,4].

As COVID-19 has become a long-term global or regional epidemic, breakthrough infections in vaccinated patients are expected, emphasising the importance of booster doses in order to prolong an effective serological immunity [5,6,7,8,9]. In October 2021, the Italian Ministry of Health (IMH) strongly recommended a third “booster” dose after at least six months from the primary vaccine cycle (two doses) in frail patients, including cancer patients. Patients with a known history of previous SARS-CoV-2 infection, determined by commercially available salivary tests, were excluded from the primary vaccination cycle or from booster dose administration in the case of infection within the last 6 months. However, it was unclear whether or how prolonged systemic cancer therapies, such as chemotherapy, targeted therapy, or immunotherapy, impact the efficacy and reactogenicity of COVID-19 vaccination [9,10,11,12,13,14,15,16,17,18,19]. Some studies have suggested lower seroconversion rates in patients who received active anticancer treatment (ACT) and particularly low responses in patients who received B-cell-depleting agents or chronic steroid therapy [8,9,10,20,21,22,23,24]. On the other hand, aberrant immune responses, including cytokine release syndrome and altered reactogenicity, have been reported in patients who received immune checkpoint inhibitors (ICIs), blocking the PD-1/PD-L1 coinhibitory axis [15,25,26,27,28,29]. Consequently, many questions remain to be addressed for long-survivor cancer patients regarding the timing of the COVID-19 booster vaccination and the likelihood of provoking unnecessary severe post-vaccine reactions or immune-related toxicity (ir-AEs), leading to treatment delay or discontinuation [30].

In order to address these concerns and provide evidence for clinical practice, between October 2021 and February 2022, we conducted an observational open-label trial aimed to identify the immunogenicity and reactogenicity of booster of the BNT162b2 vaccine in a real-world cohort of advanced lung cancer patients according to the timing of administration and/or SARS-CoV-2 infection during oncological history. As control cohorts, we considered other cancer patients (OC pts) who had completed the primary vaccination program during ACT. As an exploratory endpoint, we evaluated adverse events (AEs), clinical laboratory data and serum cytokine responses in patients undergoing ICIs treatment and COVID-19 vaccination.

## 2. Materials and Methods

### 2.1. Study Design, Eligibility and Population Cohorts

The COVALENCE study is a single-institution, open-label, multi-cohort, observational trial aimed to assess the immunological effectiveness and safety of BNT162b2 boosters in a real-world oncological population. The study prospectively enrolled patients with advanced/metastatic lung cancers (LC pts) on active ACT who had received SARS-CoV-2 primary vaccination and intended to receive or had received a BNT162b2 vaccine booster dose. Based on real-world practice, LC patients were assigned into three different cohorts, according to the timing of the third “booster” dose (BD) administration during oncological history (Appendix A):

-Cohort A: LC pts with a known history of infection with SARS-CoV-2 (determined by a previous positive salivary COVID-19 PCR test) before the primary vaccination and/or a third booster dose administration.-Cohort B: LC pts without a known history of SARS-CoV-2 infection who completed primary vaccination (2 doses) during ACT.-Cohort C: LC patients without a known history of SARS-CoV2 infection who had completed primary vaccination (2 doses) before cancer diagnosis and were candidates to receive a third dose after starting ACT.

As control cohorts, we included two cohorts of patients affected by advanced/metastatic solid cancer other than lung cancer (OC patients), without a known history of SARS-CoV-2 infection, who had completed primary vaccination and were candidates for booster dose administration. According to the prevalent ACT toxicity, OC patients were assigned to cohort D if they were receiving ACT with myelosuppressive activity (e.g., chemotherapy, cyclin-dependent kinase (CDK4/6) inhibitors), or they were included in cohort E if they were treated with ACT associated with any or low risk of myelosuppression (e.g., tyrosine kinase inhibitors (TKIs), immunotherapy (IT), antiangiogenetic drugs, others). As common eligibility criteria for cohorts A-B-D-E, patients must have received the same ACT when the primary vaccination cycle and third dose were administered. In order to allow for a long-term follow-up, at the time of the enrolment, all cancer patients must have at least a 6-month life expectancy, according to the judgment of their oncology care team.

### 2.2. Study Procedures

This study was advertised across the cancer centre, and the patients were also directly referred by their oncology care team. Written informed consent was obtained. Clinical information was abstracted from the medical record, including baseline demographics, medical history, SARS-CoV-2 exposure, cancer type and treatment history and complete blood counts obtained at the last visit before vaccination. After enrolment, the medical staff collected vaccination information and post-vaccine symptoms (vaccine reactogenicity).

Blood sampling for antibody (Ab) testing was performed at two pre-defined timepoints (T1: +30–90 days after the booster dose administration; T2: +6 months +/− 4 weeks after the booster dose). The results of the antibody testing were returned to the participants. This study was approved by the Central Ethics Committee at Istituto Spallanzani in Rome (Del. n. 487_2021).

### 2.3. Laboratory Assays

Serum quantitative antibody assays (Anti-spike IgG, anti-S IgG) were performed by Atellica IM SARS-CoV-2 IgG (COV2G), run on the Atellica IM Analyzer (Siemens Healthineers, Erlangen, Germany), at the Institutional Core Microbiology laboratory, a CLIA-certified laboratory, according to the Manufacturer’s instructions. The quantification range was between 0.05 (BAU = 1.0) and 150 U/mL (BAU = 3270). The cut-off index for a positive titre result was set at ≥1 U/mL (>21.8 BAU). As per protocol, total anti-S IgG antibody concentrations above the higher measurement limit (150 U/mL) did not trigger additional dilutions. The participants with a negative test result (<1 U/mL) were offered confirmatory testing 7–14 days later. In order to allow intercomparison with other standardised tools, thereafter, the results were converted into BAU/mL, according to the manufacturer’s conversion factor (BAU/mL = U/mL × 21.8).

Based on literature data, we also performed an exploratory panel of four cytokines (three pro-inflammatory, i.e., IL-6, IL-2R, and TNF-α, and one anti-inflammatory, i.e., IL-10) that have been reported both as significantly involved in cellular and humoral immune responses induced by the mRNA vaccine as well heavily dysregulated in post-vaccine cytokine release syndrome (CRS) [29,31,32,33]. All four cytokines were assayed in duplicate by Luminex Assays (R& D Systems, Minneapolis, MN, USA) using a Luminex xMAP system (Bio-Plex 200 array reader, Bio-Rad Laboratories, CA, USA) consisting of a multiplex biometric ELISA-based immunoassay containing dyed microspheres conjugated with a monoclonal antibody specific for a target protein. The cytokine concentrations in the plasma samples were determined from the standard curve using a five-point regression with software provided by the manufacturer (Bio-Plex Manager Software 6.0).

As there are no reference normal intervals for these analytics in cancer patients, a pre-vaccination reference value of each patient was defined as the median value of at least two previous collected samples.

### 2.4. Primary Objectives and Endpoints

The main objective of this study was to assess the long-term immunogenicity and reactogenicity of SARS-CoV-2 vaccines in lung cancer patients. The primary endpoints were (1) the rate of seropositivity for binding antibodies (IgG Anti-spike-RBD; anti-S IgG) after 30 days from the booster dose (timeframe: 30–90 days); (2) the median concentration of anti-S IgG after 6 months from the booster dose across the study cohorts and subgroups; and (3) the rate of vaccine-related reactions within 4 weeks from the third dose administration.

Local and systemic adverse effects after vaccination were assessed according to CTCAE v. 5.0. Local reactions were reported as an AE if they were ≥G3 or if they lasted longer than 48 h. Systemic reactions were reported as an AE if they lasted more than 24 h.

### 2.5. Exploratory Objective

As an exploratory scope, we analysed serum cytokine responses in patients undergoing combined ICI and COVID-19 vaccination in order to identify the risk of enhanced vaccine reactogenicity in ICI-treated patients and the risk of a higher incidence of immune-related adverse events (ir-AEs) following the booster administration. Because it is widely accepted that in patients receiving ICIs, changes in immune cell activity, leading to response or treatment failure, are evoked during early treatment administrations, we expected that 4–6 weeks after treatment starts, blood cytokine levels in the blood samples are stable and unaffected by the treatment outcome [34,35,36]. Consequently, for this exploratory objective, we considered COVALENCE participants undergoing concurrent ICI therapy and COVID-19 vaccination (cohorts A-B-C-E) with at least two pre-vaccination samples collected ≥3 months after treatment started. As there are no reference normal intervals for these analytics in cancer patients, the basal value of each patient was defined as the median value of at least two pre-vaccination assays. Post-vaccine samples were collected ≤6 weeks after booster administration. We considered as significant a 1.5-fold increase in at least two analysed cytokines compared to the pre-vaccine value (Appendix A).

### 2.6. Statistical Analysis

Descriptive statistics were performed to report the patients’ baseline characteristics, the rate of seropositivity and the prevalence of vaccine-related adverse effects by pre-specified cohort. Frequency, percentage, mean or median and standard deviation or interquartile range were reported, as appropriate. General baseline differences (age, sex, ECOG, line of therapy) across the study cohorts were tested by a Kruskal–Wallis test for continuous variables and by a chi-squared or Fisher test for categorical variables, as appropriate; the *p*-values for variables were calculated using a Monte Carlo simulation based on 2000 replicates.

Statistical differences were assessed using parametric or non-parametric statistical tests based on the variables’ distributions. A visual inspection of Q-Q plots and P-P plots was used to choose the appropriate test. Differences in the quantitative antibody titres at 6 months across the study cohorts were assessed using the Kruskal–Wallis test. Dunn’s test with Bonferroni correction was used to account for multiplicity in multiple comparisons. The Mann–Whitney test was used to assess statistical differences for continuous variables among different groups. The correlation between variables was assessed with Spearman’s rho. The relationship between age and log-transformed antibody titres was assed using univariate linear regression, and trends between different groups where evaluated using the Jonckheere–Terpstra test.

In the light of the descriptive nature of this study, exploratory analyses were performed: antibody titres were compared among different subgroups including age, sex, cancer (lung cancer and other malignancies), treatment (immunotherapy and immunosuppressive therapy), line of therapy and SARS-CoV-2 infection.

Statistical analysis was conducted using R (ver 4.1.3) and Graphpad Prism 9.4.1. All tests were two-tailed, and the threshold for statistical significance was set at *p* < 0.05.

Because of the explorative aim of this study and the lack of data from a comparable population (long-survival lung cancer patients), no pre-specified hypothesis was postulated, and no formal sample size calculation was planned. This study expected to enrol all consecutive eligible patients in a daily clinical setting.

## 3. Results

### 3.1. Patient Characteristics

Between October 2021 and February 2022, we enrolled 168 cancer patients (CPs) into this study. In total, 113 LC patients were assigned to cohort A (20 pts), cohort B (61 pts) or cohort C (32 pts); 55 OC patients were allocated to cohort D (25 pts) and to cohort E (30 pts). At the end of the pre-planned 6-month follow-up (30 September 2022), 166 patients were eligible for primary endpoint analysis (Appendix A). The demographic, cancer and therapy characteristics of the patients included in each cohort are summarised in Table 1.

Despite the limited patient sizes of our cohorts, no significant statistical difference was found in their general, clinical and demographic characteristics (Appendix A), and they were hence deemed suitable for further comparisons. Notably, the majority of patients received first-line therapy for advanced/metastatic disease. In each lung cancer cohort, more than 50.0% of the patients received immunotherapy alone or in combination with chemotherapy; moreover, four patients in cohort B discontinued immunotherapy treatment due to the end of planned cycles or immune-related AEs, achieving long-term disease control. Additionally, 73.0% of the OC in cohort E were receiving single-agent immunotherapy. The majority of patients (69.8%) had completed the primary vaccine series with the BNT162b2 vaccine (Appendix A). Throughout the follow-up period, as the Omicron variant wave spread in Italy, 40 patients (24%) experienced laboratory-confirmed SARS-CoV-2 infection before T2 evaluation, including 3 patients in cohort A who were re-infected. The spectrum of COVID-19 disease ranged from asymptomatic to mild pneumonia requiring hospitalisation, without severe complications or fatal outcomes (Appendix A).

### 3.2. Immunogenicity of the Booster Dose

After being fully vaccinated, a positive titre of S-IgG antibody (≥28.1 BAU/mL) was detected in all CP samples (168/168, 100%) collected at T1. After 6 months (T2: 6 months +/− 4 weeks), 164 patients were still seropositive (98,8%), despite 8 patients reporting a titre within the 2-fold positivity cut-off (S-Ig < 56.2 BAU/mL). Two patients had a qualitative negative result (S-IgG < 28.1 BAU/mL/mL) (Table 2). In the primary analysis between cohorts, the S-IgG concentration differed significantly only between the LC patients included in groups A-B (*p* = 0.01) and A-C (*p* = 0.02) (Figure 1A).

We next examined if patient therapy and demographic characteristics influenced the antibody response to vaccination across all the study cohorts. The median quantitative antibody concentration was lower in IO-treated CPs (1687.1 BAU/mL, interquartile range [IQR]: 437.3–3270.0) compared to non-IO treated CPs (3270.0 UI/mL, IQR: 1008.9–3270.0), with borderline statistical significance (*p* = 0.05). On the contrary, we did not find any statistical significance of median S-IgG titres when comparing CPs who received or did not receive myelosuppressive treatments (3270.0 BAU/mL vs. 2714.1 BAU/mL, *p* = 0.57) (Figure 1B,C; Appendix A).

Compared to uninfected CPs, pre-booster SARS-CoV-2 infection was associated with significantly higher antibody titres (3270.0 BAU/mL, IQR 3270.0- vs. 2174.3, IQR 553.9–3270.0; *p* = 0.004), as well as post-booster infections (3270.0 BUA/mL, IQR 3270.0–3270.0 vs. 1275.9 BAU/mL, IQR 3270.0–3270.0; *p* = <0.0001). Globally, after six months, hybrid immunisation resulted in being strongly associated with higher binding Ab titres compared to vaccine-induced seropositivity (3270.0 BAU/mL IQR: 3270.0–3270.0 vs. 376.5 BAU/mL IQR: 131.3–3270.0; *p* = <0.0001) (Figure 2A–C). Based on these preliminary findings, the patients with hybrid immunisation were not included in the subsequent analyses.

In the uninfected CPs, the median anti-S IgG titre was significantly lower in the IO-treated patients compared to the non-IO-treated patients (1732.0 BAU/mL, IQR 555.5–3270.0 vs. 757.1 BAU/mL, IQR 33.3–2863.6; *p* = 0.02) (Figure 2E). In the subgroup of LC patients, no statistically significant difference was observed in the IO-treated patients compared to the non-IO patients (1044.2 BAU/mL, IQR 323.5–3270.0 vs. 737.9, IQR 250.7–286.7; *p* = 0.43), and no difference was also observed when comparing different treatments (IO vs. CT + IO vs. TKI; *p* = 0.37) (Appendix A).

In the overall population, a medium but highly significant correlation (r = −0.31, *p* < 0.0001) was found between age and the log-transformed values of antibody titres. In the univariate regression analysis, an older age was correlated with a lower Ab titre (beta = −0.038, *p* = 0.0009), but only a small part of heterogeneity was explained by the model (r^2^ = 0.07). A weaker, but still significant, correlation (r = −0.21, *p* < 0.03) was found between age and the log-transformed values of the antibody titres in the uninfected patients. In the univariate regression analysis, the relationship between age and Ab titres was borderline significant (beta = −0.03, *p* = 0.06), explaining only a very small part of h heterogeneity (r^2^ = 0.03). In the COVID-19-naive patients, when age was categorised in three groups (<60 years, 60–69 years and ≥70 years), no statistical difference was observed (*p* = 0.018) but the evidence for a decreasing trend in Ab titres related to an increase in age was borderline significant (*p* = 0.07) (Figure 1D–F) (Appendix A).

In the uninfected CPs, no difference was observed in relation to sex (female: 1009.3 BAU/mL, IQR 5.2–3270.0 vs. male 47.51, IQR 12.02–150.00; *p* = 0.29), myelosuppressive therapy (MS therapy: 1035.7 BAU/mL, IQR 420.7–3270.0 vs. non-MS therapy: 893.8, IQR 298.8–3270.0, *p* = 0.70) or line of treatment (first line: 888.8 BAU/mL, IQR 205.8–3270.0 vs. second or subsequent line: 1456.9 BAU/mL, IQR 601.9–3270.0; *p* = 0.10) (Figure 2D–G).

### 3.3. Reactogenicity of the Booster Dose

We assessed local and systemic adverse effects after vaccination according to CTCAE v. 5.0. The majority of the participants, 51.0% (87/168 pts), reported at least one local or systemic symptom after the booster dose vaccination. The most frequent local symptom was pain at the site of injection; all local reactions were not severe (G1–G2) and recovered within one week. The most common systemic symptoms were G1–G2 malaise and/or fatigue, lasting for several weeks after vaccination (Appendix A; Table 2). No significant correlation was observed between reactogenicity and type of treatment across the study groups.

### 3.4. Cytokine Release After COVID-19 Vaccination

Among the study population, we identified a cohort of 68 IO-treated patients eligible for explorative analysis of blood cytokines. The details of concurrent ICI treatment at the time of booster vaccination are summarised in Appendix A. The cohort mainly included patients receiving ICI treatment for lung cancers (69.0%), melanoma (13.0%) and clear cell renal carcinoma (7.0%). At the time of booster vaccination, 58.8% of the patients had a confirmed radiological response from at least 1 year. Post-booster samples were collected at a median time of 10 days after booster administration.

Overall, there were no statistically significant differences in pro-inflammatory cytokine (IL2-R, IL-6, TNFα) levels between the median pre-booster and post-booster measurements. In contrast, a borderline statistically significant increment in the median concentration of IL-10 was found in the post-booster samples (median pre-BD: 0.025 pg/mL vs. post-BD: 0.350 pg/mL; *p* = 0.06) (Figure 3A). Considering a single-patient longitudinal assessment (Figure 3B), seven patients (10.3%) reported a ≥1.5-fold increase in at least two of the analysed cytokines compared to the pre-vaccine values; however, these alterations were not associated with increased reactogenicity, incidence or worsening of immune-related AEs or a modification of the cancer-response (Figure 3C; Appendix A). Additionally, 15 patients reported a ≥1.5-fold increase in only one cytokine in the post-BD samples (IL-2R: 5 pts, IL-6: 3 pts; TNF-α: 2 pts, IL-10: 7 pts); also, in these cases, we did not evidence abnormal reactogenicity.

## 4. Discussion

As COVID-19 has become a long-term global epidemic, vaccination remains critical for lung cancer patients in order to prevent cancer treatment delays, severe illness, hospitalisations and death. Both SARS-CoV-2 infection and mRNA vaccines (Pfizer/BNT162b2 and Moderna/mRNA-1273) induce a robust humoral and cellular immune response, potentially conferring protection from subsequent symptomatic infection for at least six months [18,20,31,37,38,39,40]. Multiple studies have shown that antibody titres correlate with protection at a population level, but the individual protection threshold titre remains unknown. For frail populations, such as the elderly and cancer patients, protection levels may be reduced after both vaccination and infection, and the neutralising activity of anti-spike antibodies may not be the same for all variants of the virus [9,12,18,19,23].

Methodologically, planning long-term prospective trials in cancer patients is challenging due to the risk of introducing ineluctable fundamental biases, including prognostic factors, cancer staging and treatment heterogeneity, and variation in anticancer treatments during the different timing of vaccination cycles. Therefore, in unselected cancer populations, numerous confounding variables do not allow for any realistic speculation to support pragmatic choices. Thanks to impressive advances in targeted therapies and immunotherapy, lung cancer patients represent a unique high-risk population for assessing the long-term outcomes of COVID-19 vaccination. Indeed, long-survivor LC pts are prospective candidates for receiving periodic additional vaccine doses while continuing the same anticancer treatment.

In our study, almost all patients (98.8%) generated positive titres of anti-S Ab that remain detectable at least 6–8 months following booster vaccination. As reported in healthy people, S-IgG titres progressively decay following a booster, showing a decreasing trend in the Ab concentration related to an elderly age but without any correlation with concomitant myelosuppressive ACT, such as chemotherapy or CDK4/6 inhibitors. Similar to our findings in patients with NSCLC, other studies also showed a trend in a waning antibody response after booster vaccination in elderly healthy vaccines [7,9,14]. Additionally, SARS-CoV-2 infection, whether it occurred before or after booster vaccination, seems to elicit strong hybrid immunity, resulting in higher anti-S Ab titres and further prolonged protection. Globally, these findings corroborate the suggestion that in patients with solid tumours, the immune response kinetics are more similar to those of the healthy population rather than to patients with hematologic malignancies or patients receiving immunosuppressive medications, who presented lower seroconversion rates [23,24,41,42,43,44]. The effect of immunotherapy on the vaccine response is not well-known, raising the level of hesitation around increased SARS-CoV-2 vaccine toxicity in cancer patients receiving immunotherapy. Previous studies [18,25,26,27,28] did not see a significant difference in the antibody response to boosters in patients receiving PD-1-targeted therapy compared with healthy vaccines or patients receiving other cancer therapy at the time of vaccination. On the contrary, in our study, we enrolled long-term immunotherapy responders and, after six months, we found a lower median vaccine-induced antibody titre in immunotherapy-treated patients compared to immunotherapy-unexposed patients. Although these findings can be counterintuitive at first, they may be fairly explained by considering the fundamental mechanism underlying mRNA vaccine technology [45,46]. Indeed, lipid-nanoparticle-encapsulated mRNA (mRNA-LNPs) are uptaken by antigen-presenting cells (APCs), which then traffic to the lymph nodes where they are able prime CD4^+^ T-cells. Because PD1-mediated T cell dysfunction in cancer patients leaves their tumours markedly more responsive to ICIs [27,31,34,35], we may hypothesise that vaccination can induce less robust cellular and humoral immunity in long-term immunotherapy responders, who were mainly included in our population.

In previous studies [28,29], an upregulation of pro-inflammatory cytokines after vaccination was commonly found. Also, in our exploratory analysis, one patient in ten reported a ≥ 1.5-fold increase in at least two of the analysed cytokines, without, however, experiencing systemic reactogenicity and new or worsening pre-existing ir-AEs following booster administration. Consequently, the administration of BDs of the mRNA COVID-19 vaccine was found to be safe, although transient alteration of systemic inflammation status is expected as a result of T cell priming (IL2-R, IL-6, TNF-α), followed by the deactivation (IL-10) of pro-inflammatory cytokine synthesis [29].

As this study was conducted during the pandemic, we are aware that some limitations emerged. Firstly, patients with previous SARS-Cov2 infection were identified by positive swab tests, introducing the risk of undetected infection. However, at the time of this study, clinical and molecular diagnosis (salivary PCR) was the only criteria available for physicians to decide if the patients required vaccine administration or not, reflecting real-world practice. Additionally, serological tests (such as anti-nucleocapsid antibodies) were not validated in non-experimental settings, and their use was strongly discouraged in order to compromise adherence to vaccination planning. At the same time, heterogeneity in anticancer treatments across different cancer types (including target treatments, immunotherapy and chemotherapy, as well as a combination of different strategies) does not allow for a comprehensive definition of treatment activity on the ability to generate serological immunity. Consequently, we are aware that the distinction between myelosuppressive and anticancer treatments can result arbitrarily, but also, in this case, this simplified approach reflected physician algorithms for estimating and managing the risk of infection at the time of the pandemic.

In conclusion, in our study, we saw that in all outcomes, including immunogenicity, the infectivity rate throughout the six-month period and safety, the lung cancer patients depicted a similar trend as the other patients with solid tumours as well as the general population. Consequently, the administration of booster doses of the COVID-19 mRNA vaccine in long-survivor lung cancer patients was effective and safe, reducing the risk of breakthrough infections and severe illness. Considering antibody kinetics, the planning of booster doses should be considered to enhance serological immunity, especially for elderly and long-term immunotherapy-treated patients without recent SARS-CoV-2 infection.

## Figures and Tables

**Figure 1 vaccines-13-00273-f001:**
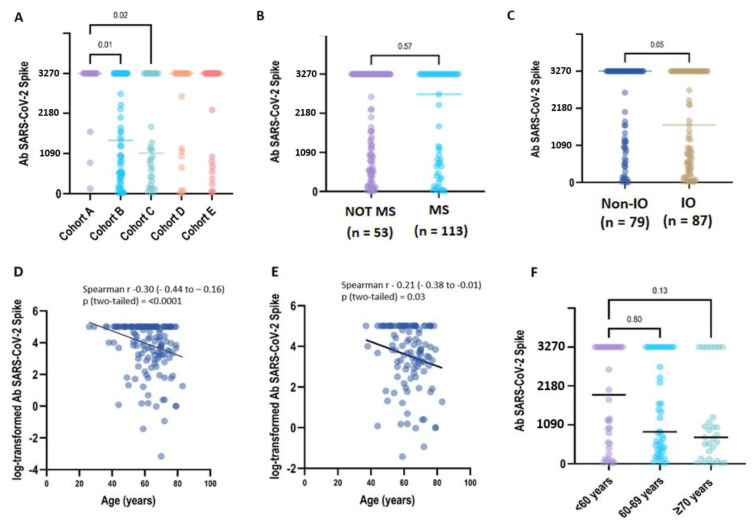
Immunogenicity of the BNT162b2 booster dose in COVALENCE participants. (**A**) SARS-CoV-2 spike IgG antibody concentration across study cohorts. Total anti-S IgG concentrations above the higher measurement limit (3270 BAU/mL) are grouped at the HML value (Appendix A). Anti-S IgG antibody titres in the plasma from patients were evaluated according to ACT myelosuppressive activity (**B**) and immunotherapy exposure (**C**). In the univariate regression analysis, older age was correlated to a lower Ab titre, either analysing the overall population (**D**) (beta = −0.03, *p* = 0.0009) or considering uninfected patients (**E**) (beta = −0.03, *p* = 0.06). In uninfected patients, when age was categorised in three groups (**F**), no statistical difference was observed (*p* = 0.018). However, the evidence for a decreasing trend in Ab titres related to age increase was borderline significant (*p* = 0.07) (Appendix A).

**Figure 2 vaccines-13-00273-f002:**
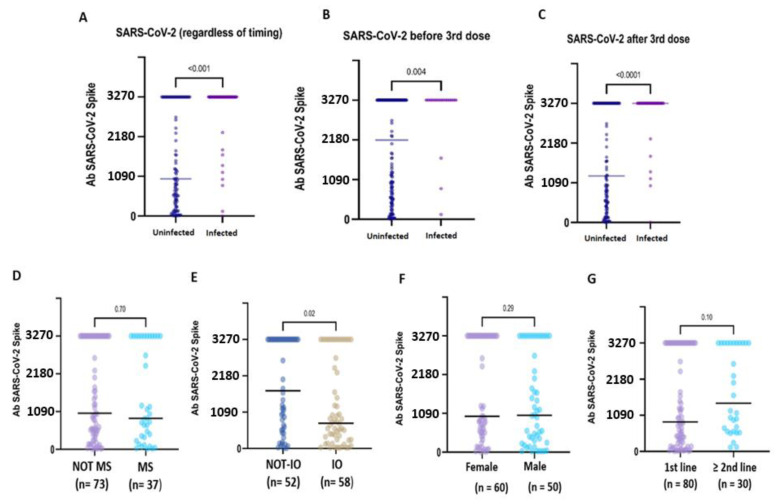
Immunogenicity of the booster dose in subgroups of hybrid-immunised and uninfected patients. Anti-S IgG antibody concentration in patients with or without a history of SARS-CoV-2 infection (**A**). The patients who developed hybrid immunisation due to SARS-CoV-2 virus infection before (**B**) or after (**C**) the booster dose administration showed very high median anti-S IgG titres compared to the uninfected patients. In the uninfected patients, no statistical difference emerged in relation to myelosuppressive ACT (**D**), sex (**F**) and the line of treatment (**G**). In contrast, the IO-treated (**E**) patients showed a significantly lower median titre compared to the non-IO patients (1732.0 BAU/mL vs. 757.1 BAU/mL; *p* = 0.02).

**Figure 3 vaccines-13-00273-f003:**
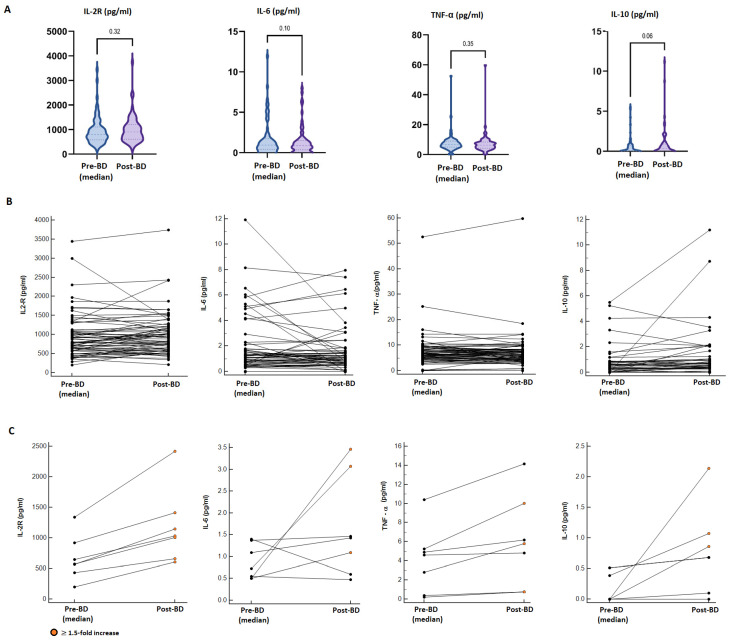
Cytokine dynamics after booster dose administration. (**A**) Comparison between the median pre-BD value of four analysed cytokines (IL2-R, IL-6, TNFα, IL-10) and the post-BD measurement. (**B**) Longitudinal plots of the cytokine concentration (pg/mL) before and after BD; lines connect paired samples, there were no statistically significant differences found by the Wilcoxon matched pairs signed rank test. (**C**) Longitudinal plots of seven patients reporting a ≥1.5-fold increase in at least two cytokines; orange dots point out ≥1.5-fold increased post-BD values.

**Table 1 vaccines-13-00273-t001:** Patients’ characteristics.

	Cohort A *n* = 20	Cohort B *n* = 61	Cohort C *n* = 32	Cohort D *n* = 25	Cohort E *n* = 30
Age, years					
Median (range)	59 (41–79)	62 (26–79)	62 (36–77)	59 (29–79)	61 (44 –83)
Sex					
Male	13 (65%)	32 (52%)	17 (53%)	6 (24%)	16 (53%)
Female	7 (35%)	29 (48%)	15 (47%)	19 (76%)	14 (47%)
ECOG					
0	17 (85%)	55 (90%)	26 (81%)	20 (75%)	27 (90%)
1	3 (15%)	6 (10%)	6 (19%)	5 (25%)	3 (10%)
Lung cancer histology					
Adenocarcinoma	16 (80%)	54 (89%)	26 (81%)	NA	NA
Squamous	-	2 (3%)	3 (9%)		
SCLC	4 (20%)	5 (8%)	2 (6%)		
Other	-	-	1 (4%)		
Other cancers (primary)					
Breast	NA	NA	NA	8 (32%)	5 (16%)
Gastrointestinal				13 (56%)	6 (20%)
Genitourinary				-	5 (16%)
Melanoma				-	9 (30%)
Head and neck				-	4 (13%)
Sarcoma and others				4 (12%)	2 (5%)
Treatment					
CT	-	1 (2%)	-	11 (44%)	-
CT + IO	5 (25%)	11 (18%)	12 (37%)	-	-
IO	7 (35%)	25 (41%)	5 (16%)	-	22 (73%)
Target therapy	8 (40%)	24 (39%)	15 (47%)	5 (20%)	-
CT + mAbs	-	-	-	9 (36%)	-
mAbs	-	-	-	-	8 (27%)
Line of therapy					
First	13 (65%)	45 (74%)	28 (87%)	13 (52%)	22 (73%)
Second	6 (30%)	9 (15%)	4 (13%)	5 (20%)	6 (20%)
≥3rd	1 (5%)	3 (5%)	-	7 (28%)	2 (7%)
EOT ^a^	-	4 (6%)	-	-	-
SARS-CoV-2 infection ^b^					
Before third dose	20 (100%)	-	-	-	-
After third dose	3 (15%)	8 (13%)	14 (44%)	7 (28%)	8 (27%)

List of abbreviations: CT = chemotherapy; IO = immunotherapy; TKI: tyrosin kinase inhibitor; mAbs: monoclonal antibodies (including antiangiogenics and anti-HER2); EOT: end of treatment. ^a^ EOT: completion of scheduled cycles, medical decision or discontinuation due to toxicity, without radiological progression. ^b^ Laboratory-confirmed infection before or after booster dose administration.

**Table 2 vaccines-13-00273-t002:** Efficacy and reactogenicity of booster doses.

	Cohort A *n* = 20	Cohort B *n* = 61	Cohort C *n* = 32	Cohort D *n* = 25	Cohort E *n* = 30	Total *N* = 166
S-IgG titre T2 ^a^						
Negative (<1 U/mL)	0/18 (0%)	2/61 (3,2%)	0/32 (0%)	0/25 (0%)	0/30 (0%)	2/166 (1.2%)
Very low (≤2 U/mL)	0/18 (0%)	4/61 (6.4%)	1/32 (3.1%)	1/25 (4.0%)	2/30 (6.6%)	8/166 (4.8%)
	Cohort A *n* = 20	Cohort B *n* = 61	Cohort C *n* = 32	Cohort D *n* = 25	Cohort E *n* = 30	Total *N* = 168
Local reactions ^b^						
Pain	4	15	7	5	7	38 (22.6%)
Erythema	1	5	1	2	2	11 (6.5%)
Swelling	0	3	1	1	0	5 (3.0%)
Total Loc React	5	23	9	8	9	54 (32.1%)
Systemic reactions ^b^						
Fatigue	1	4	2	2	0	9 (5.4%)
Malaise	3	5	3	2	0	13 (7.7%)
Chills	1	0	0	1	2	4 (2.4%)
Fever > 38 °C	2	2	0	0	0	4 (2.4%)
Others ^c^	1	1	1	0	0	3 (1.7%)
Total Syst React	8	12	6	5	2	33 (19.6%)

^a^ S-IgG titre at timepoint 2 8 after six months). ^b^ All local and systemic reactions were reordered as G1–G2. Any reaction ≥G3 was reported. ^c^ One case of G1 pericardial effusion, one case of G2 peritoneal effusion, one case of G2 thrombocytopenia.

## Data Availability

The data presented in this study are available on request from the corresponding author after approval of the local ethical committee, according to the Italian Data Protection Authority.

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
