# Peer review of "COVALENCE STUDY: Immunogenicity and Reactogenicity of a COVID-19 mRNA Vaccine in an Open-Label Cohort of Long-Survivor Patients with Metastatic Lung Cancer"

_vaccines, 2025, doi:10.3390/vaccines13030273_

Round 1

Reviewer 1 Report

Comments and Suggestions for Authors

This manuscript explores the outcomes of the COVALENCE study, focusing on participants with advanced or metastatic lung cancer. Research on immunocompromised populations, particularly those receiving immunosuppressive or cancer treatments, is scarce in the literature. This study contributes to a field lacking in existing research and is likely to interest readers. The manuscript is well-written and organised, but some points, including typos and significant figures, need to be addressed.    

Comments.
1. Study Registration: Was the COVALENCE study registered in any registry (e.g., ClinicalTrials.gov, EU Clinical Trials Register)?
If so, suggest adding the registry name and study identifier in subsection 2.1 for clarity.

2. Redundant Term: The phrase "BNT162b2 COVID-19 mRNA" is redundant.
"BNT162b2" already specifies the platform (mRNA) and target (SARS-CoV-2 spike protein, ancestral strain). Consider using just "BNT162b2" for conciseness.

3. English Style Consistency: Suggest using a consistent style of English throughout the manuscript. Currently, both US (e.g., immunization) and UK (e.g., titre) styles are used.

4. Standardised Unit (BAU/mL): The commercial immunologic assessment tool for IgG against the RBD of SARS-CoV-2 [SARS-CoV-2 IgG (sCOVG)] already uses the standardised unit BAU/mL.
Recommend using BAU/mL for intercomparison with other standardised tools. Additionally, include a statement about the conversion factor of 21.8 (BAU/mL = U/mL × 21.8).

Please also verify the instrument's name: is it "Atellica® IM" or simply "Atellica®"?
https://www.siemens-healthineers.com/en-us/laboratory-diagnostics/assays-by-diseases-conditions/infectious-disease-assays/sars-cov-2-igg-assay

See more;
Table 1 https://pmc.ncbi.nlm.nih.gov/articles/PMC9388276/
Subsection 2.1.1  https://pmc.ncbi.nlm.nih.gov/articles/PMC8378065/   5. Decimal Separator: Suggest using a period (".") as the decimal separator instead of a comma (",") to comply with journal formatting.   6. Significant Figures: Ensure consistency in significant figures across the manuscript:
- p-values: Use a consistent number of decimal places.
Currently, p-values are reported with varying decimals (e.g., p=0.06, p=0.046, p<0.0001). Suggest standardising this.
- Immunologic Outcomes: Inconsistent significant figures were noted. For uniformity, IQR 25.48 – 150 should be revised to IQR 25.48 – 150.0.
- Percentages: For instance, in Table 2.
For example, change " (6.4%)" and "(4%)" to "(6.4%)" and "(4.0%)" for consistency.  

Typos/Errors.
1. SARS-CoV-2 Notation: Across the manuscript (e.g., Lines 23, 41, 62, 84, 105, 179, 206, Figure S1, Table 1, Table S2), ensure consistent use of "SARS-CoV-2" instead of variations like "Sars-Cov2." The abbreviation requires all capital letters.

2. Line 321: Correct "BNT1272b2" to "BNT162b2."

3. Line 355: Use superscript formatting for CD4⁺.

Author Response

COVALENCE STUDY: Immunogenicity and Reactogenicity of COVID-19 mRNA 
vaccine in an open-label cohort of long-survivor patients with Metastatic Lung Cancer
Response to Reviewer X Comments
Reviewer 1
This manuscript explores the outcomes of the COVALENCE study, focusing on participants with advanced or 
metastatic lung cancer. Research on immunocompromised populations, particularly those receiving 
immunosuppressive or cancer treatments, is scarce in the literature. This study contributes to a field lacking 
in existing research and is likely to interest readers. The manuscript is well-written and organised, but some 
points, including typos and significant figures, need to be addressed. 
Response: Thank you very much for taking the time to review this manuscript. Please find the detailed 
responses below and the corresponding revisions highlighted in in the re-submitted files.
Reviewer Comments.
1. Study Registration: Was the COVALENCE study registered in any registry (e.g., ClinicalTrials.gov, EU 
Clinical Trials Register)?
If so, suggest adding the registry name and study identifier in subsection 2.1 for clarity.
Response: Thank you for pointing this out. Purely observational studies (e.g., cohort studies) do not require 
registration. Additionally, according to indication of Minister of Health, trials investigating the outcomes of 
SARS-COV2 infection and vaccination required approval by Centralized EC (Istutito Spallazani in Rome, the 
Italian referral Center for infective diseases), following a special “emergency” procedure during the 
pandemic.
2. Redundant Term: The phrase "BNT162b2 COVID-19 mRNA" is redundant.
"BNT162b2" already specifies the platform (mRNA) and target (SARS-CoV-2 spike protein, ancestral strain). 
Consider using just "BNT162b2" for conciseness.
Response: Thank you for pointing this out. Suggested changes have been updated into the manuscript.
3. English Style Consistency: Suggest using a consistent style of English throughout the manuscript. 
Currently, both US (e.g., immunization) and UK (e.g., titre) styles are used.
Response: Thank you for pointing this out. All manuscript has been updated into consistent UK style.
4. Standardised Unit (BAU/mL): The commercial immunologic assessment tool for IgG against the RBD of 
SARS-CoV-2 [SARS-CoV-2 IgG (sCOVG)] already uses the standardised unit BAU/mL.
Recommend using BAU/mL for intercomparison with other standardised tools. Additionally, include a 
statement about the conversion factor of 21.8 (BAU/mL = U/mL × 21.8).
Please also verify the instrument's name: is it or simply "Atellica®"?
Response: Thank you for your suggestion. Technical details have been updated in methods section. Titres 
results are presented in BAU/ml. 
5. Decimal Separator: Suggest using a period (".") as the decimal separator instead of a comma (",") to 
comply with journal formatting. 
Response: Thank you, correction done
6. Significant Figures: Ensure consistency in significant figures across the manuscript:
- p-values: Use a consistent number of decimal places.
Currently, p-values are reported with varying decimals (e.g., p=0.06, p=0.046, p<0.0001). Suggest 
standardising this.
- Immunologic Outcomes: Inconsistent significant figures were noted. For uniformity, IQR 25.48 – 150 
should be revised to IQR 25.48 – 150.0.
- Percentages: For instance, in Table 2. For example, change " (6.4%)" and "(4%)" to "(6.4%)" and "(4.0%)" 
for consistency. 
Response: thank you, corrections done
Typos/Errors
1. SARS-CoV-2 Notation: Across the manuscript (e.g., Lines 23, 41, 62, 84, 105, 179, 206, Figure S1, Table 1, 
Table S2), ensure consistent use of "SARS-CoV-2" instead of variations like "Sars-Cov2." The abbreviation 
requires all capital letters.
2. Line 321: Correct "BNT1272b2" to "BNT162b2."
3. Line 355: Use superscript formatting for CD4⁺.
Response: Thank you for pointing typos/errors. All notations have been corrected in revised manuscript.

Reviewer 2 Report

Comments and Suggestions for Authors

I was invited to revise the paper entitled "Immunogenicity and Reactogenicity of COVID-19 mRNA vaccine in long-survivor patients with Metastatic Lung Cancer: the COVALENCE study". It was a clinical trial aimed to evaluate long-term immunogenicity and reactogenicity of SARS-CoV-2 vaccines in lung cancer patients. 

The study was structured in 3 different cohort with active administration of vaccine:

Cohort A: LC pts who were infected and recovered by Sars-Cov-2 before primary 84 vaccination and/or third booster dose administration

Cohort B: uninfected LC pts who completed primary vaccination (2 doses) during ACT 

Cohort C: uninfected LC patients who had completed primary vaccination (2 doses)

In addition, two control groups were enrolled:

Cohort D: solid cancer other than lung cancer patients without myelosuppressive ACT

Cohort E: solid cancer other than lung cancer patients with myelosuppressive ACT

The topic is relevant and the study was well conducted.

Main observations:

- Methods lacking in:

study period and setting;

allocation procedure;

it is unknown if patients were randomized;

sampling procedure and sample size estimation;

- It seems to be a open label trial, so this information should be added in methods and in the title;

- Authors should test differences in patients baseline characteristics presented in Table 1;

- line 352: remove the link from the word "unintuitive";

- In discussion section Authors should discuss efficacy of the booster dose in the general population and after copare their results with similar studies;

- In introduction section Authors should report the vaccination schedule proposed in Italy during study period.

Author Response

COVALENCE STUDY: Immunogenicity and Reactogenicity of COVID-19 mRNA 
vaccine in an open-label cohort of long-survivor patients with Metastatic Lung Cancer
Response to Reviewer X Comments
Reviewer 2
I was invited to revise the paper entitled "Immunogenicity and Reactogenicity of COVID-19 mRNA vaccine 
in long-survivor patients with Metastatic Lung Cancer: the COVALENCE study". It was a clinical trial aimed to 
evaluate long-term immunogenicity and reactogenicity of SARS-CoV-2 vaccines in lung cancer patients. 
The study was structured in 3 different cohort with active administration of vaccine:
Cohort A: LC pts who were infected and recovered by Sars-Cov-2 before primary 84 vaccination and/or 
third booster dose administration
Cohort B: uninfected LC pts who completed primary vaccination (2 doses) during ACT 
Cohort C: uninfected LC patients who had completed primary vaccination (2 doses)
In addition, two control groups were enrolled:
Cohort D: solid cancer other than lung cancer patients without myelosuppressive ACT
Cohort E: solid cancer other than lung cancer patients with myelosuppressive ACT
The topic is relevant and the study was well conducted.
Response: Thank you very much for taking the time to review this manuscript. Please find the detailed 
responses below and the corresponding revisions highlighted in in the re-submitted files.
Reviewer Comments
Main observations:
- Methods lacking in: study period and setting; allocation procedure; it is unknown if patients were 
randomized; sampling procedure and sample size estimation;
- It seems to be a open label trial, so this information should be added in methods and in the title;
Response: Thank you for pointing this out. Clarifications for study design and sample size have been added 
into methods. Response: thank you for your observation. We specified the enrolment period in both 
abstract and introduction.
- Authors should test differences in patients baseline characteristics presented in Table 1;
Response: Thank you observation. Considering the study design (non-randomized trial, real-world setting, 
consecutive enrolment), the heterogeneity of demographic characteristic across tumour types and the 
small sizes of each cohorts, statistical analysis do not allow any conclusive or realistic consideration
- line 352: remove the link from the word "unintuitive";
Response: thank you, correction done
- In discussion section Authors should discuss efficacy of the booster dose in the general population and 
after compare their results with similar studies;
Response: thank you for your suggestion. Similarity of antibody kinetics between healthy population and 
patients with solid tumours compared to patients with hematologic malignancies are detailed in 
“discussion section”
- In introduction section Authors should report the vaccination schedule proposed in Italy during study 
period.
Response: Thank you for pointing this out. The required statement have been added into the introduction.

Reviewer 3 Report

Comments and Suggestions for Authors

Vita et al. present their findings on serological responses to SARS-CoV-2 vaccination in lung cancer patients as compared to non-lung cancer patient cohort.

The study findings can be of interest to the oncology practitioners and those studying vaccine-related responses, however there are some issues that need to be addressed:
1. The authors refer in several parts of the paper to "neutralizing antibodies", however as far as I can tell only binding antibodies are assessed. There's one place where the authors state that they measured antibodies to anti-Spike and to anti-RBD antigens, which makes it sound like they are referring to 2 different assays (one that included the full spike and another the RBD domain only) - they only describe 1 binding antibody assay in their methods though. Similarly, in a few places in text the authors discuss how their findings address "immunity", which implies that they address protective effect of the vaccine, however there's no indication that protection against either infection or disease severity was being addressed, only serological responses. 

2. It's not clear how the authors determine which patients did and which didn't experience SARS-CoV-2 infection - was it determined based on regular PCR testing? With which regularity? Were anti-nucleocapsid serological responses monitored? Without these, there's a real possibility that "uninfected" groups in fact had infected patients in them. This needs to be accounted for in both analysis and discussion. 

3. The exploratory objectives are not clearly described and therefore it's not entirely clear what is being assessed - are the authors trying to determine whether cytokine profiles correlate to vaccine adverse effects? Or whether vaccination affects cytokine profiles and that might disrupt response to treatment? 

4. There are multiple occurrences throughout the text where SARS-CoV-2 is not appropriately abbreviated.

5. There are several abbreviations that are not defined, both in text and in figure legends. Some abbreviations are inconsistently used (e.g. AB and Ab, which I presume are "antibody", but that's not actually defined)

6. Authors state that "SARS-CoV-2 might become a long-term global epidemic" - I'm assuming that they mean that it will circulate endemically. The paper might have been initially written a while ago, accounting for why the authors say that it "might" - it definitely did already start to circulate globally as an endemic virus. 

7. Would suggest that the authors state upfront over what timeframe the patients were being recruited (in both abstract and introduction), i.e. during which years of the pandemic. 

8. There are a couple of places where authors left author instructions as part of the manuscript.

9. It's not clear what's the difference between cohort D and cohort E - D is described as "anti-cancer therapy with myelosuppressive activity" and E is descried as "anti-cancer therapy with any or low myelosuppression" - both sound like patients on treatment that results in myelosuppression

10. Table 1 makes it look like none of the patients in D and E cohorts were infected prior to 3rd dose of vaccine - is it actually true or does the table need to be modified?

Comments on the Quality of English Language

The paper does require revision for clarity where English is concerned - there are some sections that are not clear, e.g. the "exploratory analysis". 

Discussion in particular is difficult to get through. 

Author Response

COVALENCE STUDY: Immunogenicity and Reactogenicity of COVID-19 mRNA 
vaccine in an open-label cohort of long-survivor patients with Metastatic Lung Cancer
Response to Reviewer X Comments
Reviewer 3
Vita et al. present their findings on serological responses to SARS-CoV-2 vaccination in lung cancer patients 
as compared to non-lung cancer patient cohort.
The study findings can be of interest to the oncology practitioners and those studying vaccine-related 
responses, however there are some issues that need to be addressed.
Response: Thank you very much for taking the time to review this manuscript. Please find the detailed 
responses below and the corresponding revisions highlighted in in the re-submitted files.
Reviewer Comments
1. The authors refer in several parts of the paper to "neutralizing antibodies", however as far as I can tell 
only binding antibodies are assessed. There's one place where the authors state that they measured 
antibodies to anti-Spike and to anti-RBD antigens, which makes it sound like they are referring to 2 
different assays (one that included the full spike and another the RBD domain only) - they only describe 1 
binding antibody assay in their methods though. Similarly, in a few places in text the authors discuss how 
their findings address "immunity", which implies that they address protective effect of the vaccine, 
however there's no indication that protection against either infection or disease severity was being 
addressed, only serological responses.
Response: Thank you for suggestions. We correct the text with “binding antibodies” and with 
“serological/humoral immunity” as appropriate
2. It's not clear how the authors determine which patients did and which didn't experience SARS-CoV-2 
infection - was it determined based on regular PCR testing? With which regularity? Were anti-nucleocapsid 
serological responses monitored? Without these, there's a real possibility that "uninfected" groups in fact 
had infected patients in them. This needs to be accounted for in both analysis and discussion. 
Response: Thank you for pointing this out. According to real-word setting and decision making process at 
the time of the study, history of SARS-CoV2 infection was determined by swab test. This issue was clarified 
in “methods” and then discussed as possible study limitation.
3. The exploratory objectives are not clearly described and therefore it's not entirely clear what is being 
assessed - are the authors trying to determine whether cytokine profiles correlate to vaccine adverse 
effects? Or whether vaccination affects cytokine profiles and that might disrupt response to treatment? 
Response: Thank you for suggestions. Clarifications have been added in the section
4. There are multiple occurrences throughout the text where SARS-CoV-2 is not appropriately abbreviated.
Response: thank you, corrections done
5. There are several abbreviations that are not defined, both in text and in figure legends. Some 
abbreviations are inconsistently used (e.g. AB and Ab, which I presume are "antibody", but that's not 
actually defined)
Response: thank you, corrections done
6. Authors state that "SARS-CoV-2 might become a long-term global epidemic" - I'm assuming that they 
mean that it will circulate endemically. The paper might have been initially written a while ago, accounting 
for why the authors say that it "might" - it definitely did already start to circulate globally as an endemic 
virus. 
Response: thank you for your observation, correction done
7. Would suggest that the authors state upfront over what timeframe the patients were being recruited (in 
both abstract and introduction), i.e. during which years of the pandemic. 
Response: thank you for your observation. We specified the enrolment period in both abstract and 
introduction.
8. There are a couple of places where authors left author instructions as part of the manuscript.
Response: thank you, corrections done
9. It's not clear what's the difference between cohort D and cohort E - D is described as "anti-cancer therapy 
with myelosuppressive activity" and E is descried as "anti-cancer therapy with any or low myelosuppression" 
- both sound like patients on treatment that results in myelosuppression
Response: thank you for your observation. Identification of myelosuppressive or not treatments have been 
detailed in “methods” section. Limitation of simplified definitions are discussed in “Conclusion” section 
10. Table 1 makes it look like none of the patients in D and E cohorts were infected prior to 3rd dose of 
vaccine - is it actually true or does the table need to be modified?
Response: thank you for your observation. We specified the absence of known history of SARS-Cov2 
infection in cohorts description

Round 2

Reviewer 2 Report

Comments and Suggestions for Authors

Authors partially addressed my previous comments.

in particular:

  • Authors stated "the heterogeneity of demographic characteristic across tumour types and the small sizes of each cohorts, statistical analysis do not allow any conclusive or realistic consideration". Baseline differences could impact the kinetics so it has to be performed.
  • Authors should discuss more in depth the impact of booster dose comparing their results with other similar studies.

Author Response

COVALENCE STUDY: Immunogenicity and Reactogenicity of COVID-19 mRNA vaccine in an open-label cohort of long-survivor patients with Metastatic Lung Cancer

Response to Reviewer X Comments

Reviewer 2 – ROUND 2

Reviewer Comments

  1. Authors stated "the heterogeneity of demographic characteristic across tumour types and the small sizes of each cohorts, statistical analysis do not allow any conclusive or realistic consideration". Baseline differences could impact the kinetics so it has to be performed.

Response: General baseline differences (age, sex, ECOG, line of therapy) across study cohorts was tested as described in “Methods” and p values have been reported in Table S2 (Supplementary)

  1. Authors should discuss more in depth the impact of booster dose comparing their results with other similar studies.

Response: Thank you for pointing this out. We discussed more extensively the key findings of our research (waning antibody response after booster in elderly and immunotherapy-treated vaccines, level of safety around increased SARS-CoV-2 vaccine toxicity in cancer patients receiving immunotherapy) and compared our results with literature background.

Round 3

Reviewer 2 Report

Comments and Suggestions for Authors

Now, it is acceptable